# Record linkage studies of primary care utilisation after release from prison: A scoping review protocol

Janine A. Cooper[1,2]*, Siobhan Murphy[1,2], Richard Kirk[3], Dermot O'Reilly[1,2], Michael Donnelly[1,2]

1 Centre for Public Health, Queen's University Belfast, Royal Hospitals Site, Belfast, United Kingdom, 2 Administrative Data Research Centre Northern Ireland (ADRC NI), Centre for Public Health, Queen's University Belfast, Royal Hospitals Site, Belfast, United Kingdom, 3 South Eastern Health and Social Care Trust, Ulster Hospital, Dundonald, United Kingdom

* j.cooper@qub.ac.uk

## Abstract

### Introduction

There is a need to improve the implementation and provision of continuity of care between prison and community in order that people who have been in prison and have a history of low engagement with services or who are vulnerable receive appropriate and timely health care and treatment. Observational studies using record linkage have investigated continuity of care after release from prison but this type of research evidence has not been synthesised.

### Objective

This paper presents a protocol designed to review record linkage studies about primary care utilisation after prison release in order to inform future research and guide service organisation and delivery towards people who are at-risk following release from prison.

### Methods

This scoping review will follow the framework by Arksey and O'Malley (5 stages) and guidance developed by the Joanna Briggs Institute (JBI). MEDLINE, EMBASE and Web of Science Core Collection will be searched (January 2012-March 2023) using terms relating to (i) 'former prisoners' and (ii) 'primary care'. The review will focus on observational studies that have investigated this topic using linked data from two or more sources. Two authors will independently screen titles and abstracts (step 1) and full publications (step 2) using predefined eligibility criteria. Data will be extracted from included publications using a piloted data charting form. This review will map the findings in this research area by methodology, key findings and gaps in research, and current evidence will be synthesised narratively given the expected considerable heterogeneity across studies.

**Funding:** Funding This work is funded by a grant awarded to the ADRC NI by the Economic and Social Research Council (ESRC) (grant number ES/W010240/1). The funders did not and will not have a role in study design, data collection and analysis, decision to publish, or preparation of the manuscript.

**Competing interests:** The authors have declared that no competing interests exist.

## Discussion

This review is part of a work programme on health in prison (Administrative Data Research Centre, Northern Ireland). This work may be used to inform future research, policy and practice. Findings will be shared with stakeholders, published in a peer-reviewed journal and presented at relevant conferences. Ethical approval is not required.

## Introduction

Prisoners often have complex health and social care needs including living with physical and mental health conditions, enduring periods of homelessness and unemployment, and experiencing addiction to drugs and alcohol [1]. In addition, the world prison population is ageing [2], with older prisoners having more chronic conditions, comorbidity, mental ill health and mobility issues than younger prisoners [3]. The United Nations (Nelson Mandela Rules) specifies that prisoners should receive the same standards of health care that are available to the general population [4]. These standards include access to necessary health care without discrimination (Rule 24), and that the organisation of appropriate health services should ensure continuity of care and treatment between prison and the community (Rule 25) [4]. The organisation and staffing of health care in prison settings differ between countries [5]. In prison settings, the frequent changing or turnover of staff and the transfer of prisoners may lead to continuity of care issues [6].

The National Health Service (NHS) includes general practice, community pharmacy, and dental and optometry services within primary care [7]. The King's Fund defines two types of continuity of care in general practice: (i) relationship continuity and (ii) management continuity [8]. The first type of continuity refers to a continuous therapeutic relationship between a GP and patient and management continuity encompasses care planning and co-ordination of care including the provision and sharing of information [8]. The transition from prison to community is a vulnerable time period among former prisoners, and there is a need for health and social care organisations to be responsive in terms of providing continuity of care between prison and the community setting, particularly for hard to reach or vulnerable prisoners, who may have had their specific health needs addressed in prison, and, post-release, require referral to appropriate health services [5]. For example, prisoners who do not tend to seek help or access services when they are living in the community may engage in treatment for substance use when they are in prison [9]. The health care needs of this population such as chronic illness, mental ill-health, substance use disorder and injury require dedicated attention during this time of transition [10].

Despite receiving continuous care and treatment during incarceration, often, the health of a prisoner may deteriorate after release and the overall effect of prison on health may be negative [11]. Among a cohort of men released from prison (with histories of injecting drug use), a prospective study reported high levels of unemployment, homelessness, physical and mental health morbidities, and illicit substance use during the two year follow up period [12]. Following the release from prison, individuals are at a higher risk of morbidity and mortality [13–16]. A recent scoping review (2011–2021) found that former prisoners have an increased risk of drug-related death during the first year after release from prison, with this risk particularly elevated in the first two weeks post-release [16]. Increasingly, research findings are pointing to an increased risk of hospitalisation after prison release. A retrospective study of emergency department visits among a cohort of former prisoners found that recently released prisoners were more likely to attend hospital than the general population for mental health disorders,

substance use disorders and ambulatory care sensitive conditions (for example, asthma, hypertension, and diabetes) [17].

Research has shown that in the United Kingdom, approximately half of prisoners do not have a general practitioner before entering custody [1]. Primary care in most health systems tend to have an encompassing remit and aim to meet the health needs of people in their local communities, from health promotion and disease prevention through to treatment, rehabilitation and palliative care [18]. Individuals who are released from prison may experience discrimination and/or other barriers to accessing health care. For example, a study of scripted telephone calls to primary care physicians (n = 250 universal health care providers) in British Columbia, Canada, to request an initial appointment for four patient scenarios; male former prisoner, male control, female former prisoner and female control, reported significantly fewer appointments offered to scenarios for patients recently released from prison compared to control individuals [19]. In this study, scenarios involving control individuals were 1.98 [95%CI 1.59–2.46] times more likely to be offered an initial family physician appointment compared with scenarios involving former prisoners [19]. A recent study identified diverse and complex health needs following release from prison and difficulties accessing health care as well as housing, employment, and social services [20]. A NAO report (2023) in England pointed to the lack of support and service planning, overall, for prison leavers [21]. For example, key prison-to-community staff handover meetings did not take place for around half of the prison leaver population even among prisoners who were released on a scheduled date and for whom there was adequate time to plan handover and community resettlement services [21]. The NAO report (2023) also pointed to a barrier to effective service access and receipt by prison leavers related to inefficient information sharing between health care providers and criminal justice professionals [21].

A systematic review of measured continuity of care to all patient groups by doctors in any setting (search 1996–2017), found that higher continuity of care was associated with reduced mortality [22]. There is limited evidence in terms of interventions to improve health services contact and outcomes in the transition from the prison to community setting. A systematic review of prisoners with mental health conditions found that interventions during the transition from prison to community improved contact with mental health care and other health services after release [23]. However, there was some evidence to suggest that interventions during this transition period may be associated with an increase in the number of individuals returning to prison (for example, as a result of increased monitoring) [23]. The interventions included in the review focused on different stages of release from prison and had differing content, therefore further studies are warranted in this area [23]. A recent scoping review of all study designs investigating the continuity of opioid substitution treatment between prison and community settings in Indonesia, Malaysian and Vietnam (search 2011–2021) was limited by a low number of studies and a lack of generalisability [24]. However, the review found three emergent themes: (i) facilitators of post-release continuity of care included guidelines, staff training and family support, (ii) barriers of post-release continuity of care included lack of trained staff and a high turnover of staff, lack of coordination between prison, community health centres and government departments, lack of data management between prison and community, limited availability/challenges accessing care and side-effects or concerns about treatment, and (iii) therapeutic considerations supported post-release continuity of care [24].

## Purpose of review

The purpose of this scoping review is to identify, map and summarise observational research using record-linkage studies about primary care after release from prison. The transition from

prison to community settings is a vulnerable period among former prisoners and, as such, this review will focus on the response by primary care (all detailed primary care search terms and concepts used in this review are provided in the method section below). A systematic review of empirical research on primary care contact (or use) and health related outcomes during the transitional time period after prison release is warranted to improve understanding about the needs of potentially vulnerable or at-risk individuals. For example, a population-based retrospective cohort study used linked data sources (provincial prison data and health administrative data) to investigate primary care attachment (i.e. use of primary care during the study periods) during the two years before incarceration and two years following prison release [25]. The study found significantly lower attachment to primary care in incarcerated individuals, both before admission to prison and after release, compared to the general population [25]. This review will focus on the methodologies used in record linkage studies to summarise study characteristics, designs, outcomes, findings and gaps in knowledge in order to best guide future research. A synthesis of the evidence about individuals' contacts with primary care after prison release will be applied to illustrate ways in which to increase and improve continuity of care, inform future research in this area and guide practitioners to target interventions towards those people most at-risk following prison release.

This review is part of a work programme on health in prison that is being undertaken by the Administrative Data Research Centre, Northern Ireland (ADRC NI) and may be of interest to a range of stakeholders including primary care healthcare teams, and other health and social care providers, and prisons.

## Methods

This scoping review will follow the framework by Arksey and O'Malley (5 stages) and guidance developed by the Joanna Briggs Institute (JBI) [26, 27]. The Preferred Reporting Items for Systematic reviews and Meta-Analyses extension for Scoping Reviews (PRISMA-ScR) checklist and guidance has been used to inform this protocol (S1 Checklist) and will be used to report this review [28]. The completed PRISMA checklist will be available as an appendix with the full review.

### Stage 1: Identifying the research questions

This scoping review will address the following research questions:

1. What is the scope of the research literature on record linkage studies about primary care after prison release?

2. What methodologies are reported in these studies?

3. What are the findings in relation to primary care contact by people released from prison (including any hand-over arrangements and accessing and using primary care) and any reported health or prison related outcomes?

4. Where are the knowledge gaps in this area?

### Stage 2: Identifying relevant studies

Searches will be performed on three literature databases (MEDLINE, EMBASE and Web of Science Core Collection) from January 2012 to March 2023 using keywords and index headings. To summarise the most recent evidence in this field of research, a start date of 2012 was chosen for this scoping review. The separate searches will be modified for each database as

required. A search strategy has been developed for MEDLINE which includes terms (and variants) relating to two areas 'former prisoners' and 'primary care' (provided in S1 Appendix). The search strategy was developed through discussions between JAC and MD. This strategy was also reviewed by the Subject Librarian for the School of Medicine, Dentistry and Biomedical Sciences in Queen's University Belfast. Using this search strategy has identified at least 185 potentially eligible papers in MEDLINE (S1 Appendix). Further search strategies, for EMBASE and Web of Science Core Collection, will be developed by JAC and MD and will be provided with the full review. The database searches will be performed by JAC, and the results will be combined in a reference manager (duplicate results will be removed). Due to resources for translation, the search strategy will be limited to publications available in English. Reference lists of included studies will be screened for further publications. The review will not include a search of the grey literature.

### Stage 3: Study selection

Inclusion and exclusion criteria will incorporate population, concept and context (PCC) and will be used to determine the eligibility of publications. Titles and abstracts will be independently screened by two authors (step 1). If the publication appears to meet the inclusion criteria following step 1, or there is any uncertainty over inclusion, the publication will be screened in full (step 2). All full publications will be independently screened by two authors. If any disagreements arise between the two authors during step 2, the publication will be discussed with a third author. Corresponding authors will be contacted for additional information to determine eligibility, if required.

### Eligibility criteria

**Population.**   Studies among adults released from prison into the community will be eligible for inclusion. There will be no exclusions on time periods after release from prison, but where possible this will be recorded in the data charting form.

**Concept.**   Studies addressing any contact with primary care health services after release from prison will be eligible for inclusion. As part of this selection process, the term 'primary care contact' will involve all types of health services provided within the general practice setting (for example, in-person and telephone consultations, home visits, clinic and treatment appointments). This will also include similar terminology used for these services, such family health, family physician, primary care physician and nurse practitioner etc. Contact with other primary care services such as a community pharmacy, or dental and optometry services will also be eligible for inclusion. All observational studies (i.e. cohort, case-control and cross-sectional studies) using linked data from two or more sources will be eligible for inclusion.

**Context.**   All geographical locations will be eligible for inclusion. Only research from peer-reviewed journal articles will included. All sources of data other than peer-reviewed academic journal papers, for example, conference abstracts, editorials, commentaries and letters will be excluded.

### Stage 4: Charting the data

Data will be charted from included publications. As part of this protocol development, a data charting form has been piloted by two authors (S2 Appendix). In the review, data charting will be performed independently by two authors. Initially, after charting the data for a sample of the included studies, one author (JAC) will check that information is being extracted consistently by the two authors and there will be a discussion among authors whether any further modifications are required on the form. Following this accuracy check, data charting will be

completed for all remaining included studies. If any disagreements arise in the information extracted from publications, the publication will be discussed with a third author. Corresponding authors will be contacted for additional information, if required.

## Stage 5: Collating, summarising and reporting the results

The search results and study selection process will be presented in a flow diagram in the full review. This review will map the findings in this research area by methodology, key findings and gaps in research, and current evidence will be synthesised narratively as considerable heterogeneity across studies is expected. Summary tables will be provided (for example, study characteristics of included studies and summary of findings etc.). Also, study findings will be organised by country and type of health system.

## Patient and public involvement

RK is the Clinical Director of Healthcare in Prison in Northern Ireland. No additional patient and public involvement.

## Discussion

It is hoped that by synthesising the evidence about individuals' contacts with primary care after prison release, this review may provide an opportunity to increase and improve continuity of care for those most at-risk following prison release, by helping to inform future research in this area. This review will focus on observational studies that have investigated this topic using linked data. A separate review of trials and quasi-experimental research is warranted. The search strategy will focus on 'former' incarceration and its variants in order to increase specificity. The use of such terms may pose a possible limitation as potentially some publications may be missed due to reduced sensitivity. However, the focus of the review will be primary care contact after release from prison and therefore authors feel this decision is justified. The review will not include a grey literature search and due to resources for translation, the search strategy will be limited to publications available in English which may limit the interpretation of the findings. Given the exploratory nature and breadth of a scoping review it may mean that changes to the protocol could be required as new information becomes available throughout the review process. Any changes in the full review will be documented and the rationale for these changes will be provided. Ethical approval is not required as this work will summarise publicly available sources of research. The full review will be disseminated as academic research and will be published in a peer-reviewed journal and presented at relevant conferences.

## Supporting information

**S1 Checklist. PRISMA-ScR checklist.**
(DOCX)

**S1 Appendix. Search strategy for MEDLINE.**
(DOCX)

**S2 Appendix. Data charting form.**
(DOCX)

## Acknowledgments

Thanks to Richard Fallis, Subject Librarian, School of Medicine, Dentistry and Biomedical Sciences, Queen's University Belfast, for reviewing the search strategy and for providing guidance in the use of reference management software.

## Author Contributions

**Conceptualization:** Janine A. Cooper, Richard Kirk, Dermot O'Reilly, Michael Donnelly.

**Funding acquisition:** Dermot O'Reilly, Michael Donnelly.

**Methodology:** Janine A. Cooper, Michael Donnelly.

**Project administration:** Janine A. Cooper, Michael Donnelly.

**Validation:** Janine A. Cooper, Siobhan Murphy.

**Writing – original draft:** Janine A. Cooper.

**Writing – review & editing:** Janine A. Cooper, Siobhan Murphy, Richard Kirk, Dermot O'Reilly, Michael Donnelly.

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
