## [Decision Letter · Decision Letter 0]

5 Jun 2023

PONE-D-23-08846Record linkage studies of primary care utilisation after release from prison: a scoping review protocolPLOS ONE

Dear Dr. Cooper,

I hope this email finds you well. On behalf of PLOS ONE, I would like to apologise for the delay in the decision-making process on your study protocol. We had to establish contact with a reviewer to clarify their review, which had inevitably prolonged the review process.

We have now completed this process and I am pleased to inform you that we would like to accept your study protocol, subject to a minor revision. You will see that your submission had been reviewed by two reviewers. When responding to the reviewers’ comments, we would like to ask you to ignore the recommended rejection by Reviewer 1. The reviewer was not aware that PLOS ONE accepts study protocols for publications. Subsequent correspondences with this reviewer established that they were happy to accept the publication of this study protocol, in line with the journal’s policy.

Thank you for considering PLOS ONE for this study protocol. We look forward to receiving your revised manuscript.

Kind regards,

Dr Nasrul Ismail

Academic Editor

PLOS ONE

Reviewers' comments:

Reviewer's Responses to Questions

**Comments to the Author**

1. Does the manuscript provide a valid rationale for the proposed study, with clearly identified and justified research questions?

Reviewer #1: No

Reviewer #2: Yes

2. Is the protocol technically sound and planned in a manner that will lead to a meaningful outcome and allow testing the stated hypotheses?

Reviewer #1: No

Reviewer #2: Partly

3. Is the methodology feasible and described in sufficient detail to allow the work to be replicable?

Reviewer #1: No

Reviewer #2: Yes

4. Have the authors described where all data underlying the findings will be made available when the study is complete?

Reviewer #1: Yes

Reviewer #2: Yes

5. Is the manuscript presented in an intelligible fashion and written in standard English?

Reviewer #1: No

Reviewer #2: Yes

6. Review Comments to the Author

You may also provide optional suggestions and comments to authors that they might find helpful in planning their study.

Reviewer #1: Grey literature is an important source of information in the field of prison health. I would recommend the authors to add it to their methodology.

Reviewer #2: This will be a useful scoping review for a key moment in the journey of prison residents as they transition to community care - you have highlighted this well. A few changes are suggested below:

Lines 110-111 - please expand the list of barriers to accessing primary health care for individuals released from prison. For example, in England, there can be issues with community GPs accessing prison health care records (though I believe this has recently improved). There can also be a lack of appropriate handover from prison to community services, particularly if prison residents are released at short notice.

Line 217 - I believe you should include experimental studies as well as observational studies. You state in your introduction that the review will be a "synthesis of best available evidence about individuals' contact with primary care after prison release" in order to "illustrate ways in which to increase and improve continuity of care". By excluding experimental evidence, you are potentially excluding the highest quality evidence of interventions which would improve this continuity of care.

7. PLOS authors have the option to publish the peer review history of their article (what does this mean?). If published, this will include your full peer review and any attached files.

Reviewer #1: No

Reviewer #2: **Yes: **Luke Johnson

---

## [Author Response · Author response to Decision Letter 0]

26 Jun 2023

Full Title: Record linkage studies of primary care utilisation after release from prison: a scoping review protocol

Manuscript Number: PONE-D-23-08846

Date: 26.06.2023

Comment: The authors would like to thank the Editor and reviewers for taking the time to consider this scoping review protocol. Responses to reviewer comments are provided below in red text. Where required, reviewers are directed to the relevant page numbers (in the manuscript containing track changes) for any edits made to the manuscript.

Please note that we have updated reference 16 (page 15). “Cooper JA, Onyeka I, Cardwell C, Paterson E, Kirk R, O'Reilly D, Donnelly M. Record linkage studies of drug-related deaths among adults who were released from prison to the community: a scoping review. BMC Public Health. 2023 May 5;23(1):826. doi: 10.1186/s12889-023-15673-0. PMID: 37147595; PMCID: PMC10161544.”

6. Review Comments to the Author

You may also provide optional suggestions and comments to authors that they might find helpful in planning their study.

Reviewer #1: Grey literature is an important source of information in the field of prison health. I would recommend the authors to add it to their methodology.

Response: Grey literature has been omitted for pragmatic reasons, primarily time considerations. We have noted the omission of grey literature as a potential limitation of the review. The authors’ state in the Methods section (page 9 lines 201-202) and Discussion section (page 12 lines 265-267) that the review will not include a search of the grey literature. 

Reviewer #2: This will be a useful scoping review for a key moment in the journey of prison residents as they transition to community care - you have highlighted this well. A few changes are suggested below:

Lines 110-111 - please expand the list of barriers to accessing primary health care for individuals released from prison. For example, in England, there can be issues with community GPs accessing prison health care records (though I believe this has recently improved). There can also be a lack of appropriate handover from prison to community services, particularly if prison residents are released at short notice.

Response: We thank the reviewer for this comment. 

Text removed “based on the disclosure of recent incarceration” (page 5 line 111). We have added some supporting text (page 6 lines 118-126) and publications (page 16 lines 367-374). 

“A recent study identified diverse and complex health needs following release from prison and difficulties accessing health care as well as housing, employment, and social services [Hu et al. 2020]. A NAO report (2023) in England pointed to the lack of support and service planning, overall, for prison leavers [NAO 2023]. For example, key prison-to-community staff handover meetings did not take place for around half of the prison leaver population even among prisoners who were released on a scheduled date and for whom there was adequate time to plan handover and community resettlement services [NAO 2023]. The NAO report (2023) also pointed to a barrier to effective service access and receipt by prison leavers related to inefficient information sharing between health care providers and criminal justice professionals [NAO 2023].”

“Hu C, Jurgutis J, Edwards D, O'Shea T, Regenstreif L, Bodkin C, Amster E, Kouyoumdjian FG. "When you first walk out the gates…where do [you] go?": Barriers and opportunities to achieving continuity of health care at the time of release from a provincial jail in Ontario. PLoS One. 2020 Apr 10;15(4):e0231211. doi: 10.1371/journal.pone.0231211. PMID: 32275680; PMCID: PMC7147766.”

“NAO, National Audit Office. Report. Improving resettlement support for prison leavers to reduce reoffending. Ministry of Justice, HM Prison & Probation Service. Session 2022-23, 12 May 2023 HC 1282. National Audit Office (NAO) copyright.”

Reference formatting here is for information only, the reference numbers have been updated in the manuscript containing track changes.

Line 217 - I believe you should include experimental studies as well as observational studies. You state in your introduction that the review will be a "synthesis of best available evidence about individuals' contact with primary care after prison release" in order to "illustrate ways in which to increase and improve continuity of care". By excluding experimental evidence, you are potentially excluding the highest quality evidence of interventions which would improve this continuity of care.

Response: The point about research evidence based on experimental studies is well-made. However, the proposed review is one of a series of planned reviews that will be used to inform the record-linkage type of research about the health care of released prisoners that will be undertaken by our ESRC-funded Administrative Data Research Centre (ADRC). Thus, the review will focus on observational record-linkage studies in keeping with the remit of the ADRCs in each UK nation. In the review, we will report the scope of the research literature on record linkage studies about primary care after prison release, but we will also summarise, compare and comment on methodologies used to report this research. We will include in the Discussion section that trials would benefit from a separate review. We have added text under Discussion on pages 11-12 lines 260-261.

“This review will focus on observational studies that have investigated this topic using linked data. A separate review of trials and quasi-experimental research is warranted.”

We have removed ‘best available’ from text on page 7 line 161 and page 11 line 257. 

Date: 26.06.2023

---

## [Decision Letter · Decision Letter 1]

14 Jul 2023

Record linkage studies of primary care utilisation after release from prison: a scoping review protocol

PONE-D-23-08846R1

Dear Dr. Cooper,

I am pleased to inform you that your manuscript has been judged scientifically suitable for publication and will be formally accepted for publication once it meets all outstanding technical requirements.

Kind regards,

Dr Nasrul Ismail

Academic Editor

PLOS ONE

Additional Editor Comments (optional):

N/A

Reviewers' comments:

Reviewer's Responses to Questions

**Comments to the Author**

1. Does the manuscript provide a valid rationale for the proposed study, with clearly identified and justified research questions?

Reviewer #2: Yes

2. Is the protocol technically sound and planned in a manner that will lead to a meaningful outcome and allow testing the stated hypotheses?

Reviewer #2: Yes

3. Is the methodology feasible and described in sufficient detail to allow the work to be replicable?

Reviewer #2: Yes

4. Have the authors described where all data underlying the findings will be made available when the study is complete?

Reviewer #2: Yes

5. Is the manuscript presented in an intelligible fashion and written in standard English?

Reviewer #2: Yes

6. Review Comments to the Author

You may also provide optional suggestions and comments to authors that they might find helpful in planning their study.

Reviewer #2: Thanks for your response. I have no further suggestions. All the best with your scoping review and ongoing research.

7. PLOS authors have the option to publish the peer review history of their article (what does this mean?). If published, this will include your full peer review and any attached files.

Reviewer #2: **Yes: **Dr Luke Johnson

---

## [Editor Report · Acceptance letter]

16 Aug 2023

PONE-D-23-08846R1 

Record linkage studies of primary care utilisation after release from prison: a scoping review protocol 

Dear Dr. Cooper:

I'm pleased to inform you that your manuscript has been deemed suitable for publication in PLOS ONE. Congratulations! Your manuscript is now with our production department. 

Kind regards, 

on behalf of

Dr. Nasrul Ismail 

Academic Editor

PLOS ONE